# Meta-Analysis of the Effects of Organic Chromium Supplementation on the Growth Performance and Carcass Quality of Weaned and Growing-Finishing Pigs

**DOI:** 10.3390/ani13122014

**Published:** 2023-06-16

**Authors:** Tao He, Chunbo Wei, Xiuwei Lin, Baoyin Wang, Guoan Yin

**Affiliations:** 1Key Laboratory of Low-Carbon Green Agriculture in Northeastern China, Ministry of Agriculture and Rural Affairs P. R. China, Department of Animal Science, College of Animal Science and Veterinary Medicine, Heilongjiang Bayi Agricultural University, Daqing 163319, China; ht960704@163.com (T.H.); 13630986334@163.com (B.W.); guoanyin@foxmail.com (G.Y.); 2Branch of Animal Husbandry and Veterinary of Heilongjiang Academy of Agricultural Sciences, Qiqihar 161000, China; lxw960521@163.com

**Keywords:** pigs, exogenous organic chromium, growth performance, carcass quality, meta-analysis

## Abstract

**Simple Summary:**

During pig production, organic chromium is usually added to the feed to supplement the chromium needed for pig growth and metabolism. However, different types of organic chromium have different effects on the addition period and dosage. Therefore, the purpose of this study is to analyze the effect of adding organic chromium in pig feed by means of meta-analysis. The results show that organic chromium can improve the average daily gain and average daily intake of pigs. At the same time, the effect of adding in the weaning stage is better than that in the growing and finishing stage.

**Abstract:**

Many factors influence the effects of exogenous organic chromium (EO-Cr) on the growth performance and carcass qualities of weaned and growing-finishing pigs, such as pig growth stages, types of EO-Cr, period of supplementation, and farm management. However, it is challenging to comprehensively consider all factors in one study. To solve this problem, we searched all relative literature published from 1 January 2000 to 1 January 2023, to systematically analyze and review the effects of EO-Cr on pig growth performance and carcass qualities via meta-analysis. Thirty-five papers were filtered and analyzed, which involved 4366 pigs. The results showed that, for weaned piglets, EO-Cr diets significantly increased the average daily gain (ADG, *p* < 0.001) and average daily feed intake (ADFI, *p* = 0.022) but reduced the feed–gain ratio (*p* = 0.004). In addition, for growing-finishing pigs, EO-Cr supplementation significantly increased the ADG (*p* < 0.001), carcass lean ratio (*p* = 0.020), and loin muscle area (*p* < 0.001), but had no significant effect on the ADFI (*p* = 0.071), feed–gain ratio (*p* = 0.692), dressing percent (*p* = 0.989), or back fat thickness (*p* = 0.142). Moreover, the effect of EO-Cr was greater in weaned piglets than in growing-finishing pigs. In terms of the dose effect of the supplement, chromium nicotinate is the most suitable EO-Cr type for weaned piglets with an optimal dosage range of 0.125–0.150 mg/kg. On the other hand, chromium picolinate is the most suitable EO-Cr type for growing-finishing pigs with an optimal dosage range of 0.250–0.300 mg/kg. In conclusion, EO-Cr supplementation is beneficial for enhancing the growth performance and carcass qualities of both weaned and growing-finishing pigs.

## 1. Introduction

Chromium (Cr) has long been considered a toxic and carcinogenic substance and not regarded as a useful component of animal feed [1]. However, many studies have proved that Cr is a vital trace element for maintaining normal physiological activity in animals [2]. Although Cr demand is small, it plays an essential role in sugar, fat, protein metabolism, and nucleic acid synthesis in animals [3]. In addition, low levels of dietary Cr can lead to anorexia, weight loss, reproductive disorders, and an abnormal rise of blood lipids in both humans and animals [4]. Cr is also a potent antioxidant that can prevent heat-stress-induced lipid peroxidation and combat the adverse effects of heat stress in animals [5]. As animals cannot spontaneously synthesize Cr, exogenous Cr should be added to their diets to meet their Cr requirements [6]. 

Inorganic chromium is highly toxic and should not be added directly to the diet. Instead, organic chromium, which is composed of compounds similar to glucose tolerance factor (GTF), such as chromium (III) nitrate, chromium yeast, chromium picolinate, and chromium protein, is recommended. Organic chromium is commonly used as an anti-stress additive in animal feeds. Although it is present in regular diets, its content is typically low. Both China and the United States National Research Council (NRC) recommend adding an appropriate amount of chromium in animal production; however, the specific amount is not clearly specified [7]. However, the cumulative effect of Cr is closely related to the duration of Cr supplementation. In the early stage of animal growth, the duration of Cr supplementation is short, so the effect is negligible. The results of research on the effect of exogenous organic chromium (EO-Cr) on pig growth are inconsistent; no studies have analyzed the respective advantages and disadvantages of these various forms, and it remains unclear how EO-Cr influences pig growth during different physiological periods. Therefore, we conduct a meta-analysis of five common forms of EO-Cr added to pig diets: chromium nicotinate, yeast chromium, chromium methionine, chromium picolinate, and chromium propionate. This study hopes to reveal the effect of EO-Cr on the growth performance and carcass quality of weaned and growing-finishing pigs, as well as the optimal amount and form of EO-Cr through practical evaluation.

## 2. Materials and Methods

### 2.1. Meta-Analysis

This meta-analysis is reported according to the PRISMA 2020 statement: An updated guideline for reporting systematic reviews [8].

#### 2.1.1. Search Strategies

We searched and collected relevant literature on the effects of five forms of EO-Cr (chromium nicotinate, yeast chromium, chromium methionine, chromium picolinate, and chromium propionate) on pig performance and carcass qualities published in CNKI (China; https://www.cnki.net; accessed on 25 February 2023), CQVIP (China; https://vpcs.fanyu.com; accessed on 25 February 2023), Wan Fang (China; https://www.wanfangdata.com.cn; accessed on 25 February 2023), Web of Science (https://www.webofscience.com; accessed on 25 February 2023), ScienceDirect (https://www.sciencedirect.com; accessed on 25 February 2023), and PubMed (https://pubmed.ncbi.nlm.nih.gov; accessed on 25 February 2023) databases between 1 January 2000 and 1 January2023. The detailed search strategies and findings are shown in Table A1 of Appendix A.

#### 2.1.2. Selection Criteria and Procedure

The inclusion criteria were as follows: (1) studies must include both a control group (feeding an essential diet) and an experimental group (EO-Cr supplementation); (2) these studies reported the same outcomes of feeding chromium; (3) the category name and addition amount of EO-Cr should be specified; (4) the studies should be randomized controlled trials; (5) the studies must be published in peer-reviewed journals; and (6) the studies must specify whether the feeding stage was the weaned or growing-finishing stage.

The exclusion criteria were as follows: (1) the experiments did not include a control group; (2) the study did not investigate pig production performance or carcass quality; (3) the types and dosage of EO-Cr were not specified; (4) the study was a review paper. 

The following information was retrieved from each paper included in this study: author information (first author, publication year); breeds of pig; growth stage (weaned pig or growing-finishing pig); EO-Cr types (chromium nicotinate, yeast chromium, chromium methionine, chromium picolinate, or chromium propionate); feeding amount; sample size of the selected group; period of supplementation (days of continuous supplementation of organic chromium additives); and production performance indicators, such as average daily gain (ADG), average daily feed intake (ADFI), feed gain ratio (F/G), carcass quality indexes (dressing percent, carcass lean ratio, eye muscle area, and back fat thickness), and their corresponding standard deviations (SD). If any literature did not provide SD values, weighting was performed by the inverse of the variance in a hierarchical effects model that included robust variance estimation. Each variance comparison between the EO-Cr group and control treatments was calculated as the square of the pooled SD. The SD for the EO-Cr group and control for each comparison was calculated from the reported standard error of mean (SEM) such that *n* is the number of experimental units [9].

#### 2.1.3. Research Quality Assessment

Two researchers (T. He and X. Lin) conducted an independent quality assessment study according to the joint trial report standard statement [10] and the assessment bias risk tool [11]. The evaluation included sequence generation, allocation concealment, blinding of participants and personnel, incomplete result data, selective reporting, and other biases. Differences were adjusted by a third researcher (B. Wang).

#### 2.1.4. Effect and Heterogeneity

The standardized mean difference (SMD) method was used in this study because the data type in the literature is a continuous variable. In order to reduce the heterogeneity interference and maintain the authenticity of the results, this study uses the random effect model to calculate the confidence interval of the standardized mean difference (SMD) and the corresponding 95% Cl. The effect was reflected through the combined effect amount SMD and the 95% confidence interval (CI). SMD is the difference in mean (or proportion or grade) between two groups divided by the standard deviation, where a 95% CI that includes a zero value or intersects with the invalid line in the forest map indicates no significant difference between the test group and the control group. The significance of the regulatory effect is determined by the *p*-value, and a *p*-value < 0.05 indicates that the degree of regulation of this effect is significant. The heterogeneity was determined according to the *Q*-value, *p*-value, and I^2^, where *p*-value < 0.05 and I^2^ > 50% is regarded as significant heterogeneity. We conducted subgroup analyses and meta-regression to explore the potential sources of heterogeneity. These subgroups were as follows: (1) Weaned piglets and growing-finishing pigs; (2) Weaned piglets are divided into chromium picolinate, chromium yeast, and chromium nicotinate according to different types of organic chromium; and (3) Growing-finishing pigs are divided into chromium picolinate, chromium propionate, chromium methionine, and chromium yeast. Sources of heterogeneity were explored through meta-regression analysis. Egger’s test and the fail-safe N loss factor were used to test the publication bias, where Egger’s *p* < 0.05 indicated a significant publication bias in this study. Finally, in order to help determine the optimal amount of organic chromium, we consider the amount of organic chromium added and the result of SMD to draw the dose curve.

### 2.2. Fuzzy Comprehensive Evaluation

Fuzzy comprehensive evaluation refers to the evaluation of fuzzy and uncertain problems determined by multiple factors or standards by using the grade index and weight of the evaluation factors to evaluate each factor, respectively, and the comprehensive description of each factor by using the principle of membership degree, and the establishment of a discrimination model to obtain the grade result of the evaluation object. The design of the evaluation index system is the basis of the effectiveness of the fuzzy comprehensive evaluation method. Its basic method is to decompose the objectives layer by layer and form the index system according to certain standards.

First, we determined the fuzzy evaluation factors. The EO-Cr evaluation factor set was as follows: {U_1_ daily gain, U_2_ daily feed intake, U_3_ feed gain ratio, U_4_ dressing percent, U_5_ carcass lean ratio, U_6_ eye muscle area, U_7_ back fat thickness}. According to the numerical characteristics of indicators, when there are positive indicators with higher values as the better, select the membership function (1) for calculation. When there are negative indicators with lower values as the better, select the membership function (2) for calculation.
(1)$$U_{1},U_{2},U_{4},U_{5},U_{6}=\{\begin{array}{c} \begin{array}{cc} 1 & X_{i}\geq X_{\mathrm{Max}} \end{array}\\ \begin{array}{cc} \frac{X_{i}}{X_{\mathrm{Max}}} & X_{i}< \end{array}{\;X}_{\mathrm{Max}} \end{array}$$
(2)$$U_{3},U_{7}=\{\begin{array}{c} \begin{array}{cc} 1\; & X_{i}=X_{\min} \end{array}\\ \begin{array}{cc} \frac{X_{\min}}{X_{i}}(\mathrm{No}\;\mathrm{negative}) & \frac{X_{i}}{X_{\min}}(\mathrm{Negative})X_{i}>X_{\min} \end{array} \end{array}$$

Here, X_Max_ represents the maximum value under this index, i represents the type of EO-Cr under the index, X_min_ represents the minimum value under the index, and X_i_ represents the optimal amount of EO-Cr effect value the index.

Through the membership functions in Formulas (1) and (2), a reasonable amount of data for different forms of EO-Cr in each index was substituted for calculation. Finally, the membership degree was added, and the highest score indicated the optimal form of EO-Cr.

### 2.3. Statistical Analysis

Comprehensive meta-analysis V3 software analyzes the effect value, weight, variance, and vast effect quantity of each study included in the meta-analysis. The fuzzy comprehensive evaluation was conducted in Excel. To combine the effect amount with the corresponding added dose using the ggplot2 package (https://ggplot2.tidyverse.org; accessed on 10 March 2023) in R version 4.2.2, a curve fitting graph can be drawn to analyze the optimal amount of EO-Cr added.

## 3. Results

### 3.1. Included Literature

A total of 5977 published studies were retrieved using the aforementioned methods. After screening and eliminating duplicate studies based on the inclusion and proposed criteria mentioned earlier, we selected 35 studies for this meta-analysis [12,13,14,15,16,17,18,19,20,21,22,23,24,25,26,27,28,29,30,31,32,33,34,35,36,37,38,39,40,41,42,43,44,45,46], which involved 4366 pigs. The details of these studies can be found in Table A2 of Appendix A.

### 3.2. Research Quality Evaluation Results

After evaluating the research quality of the 35 articles included in this study, as depicted in Figure 1, it was determined that the primary biases were of low risk, with a minor amount of high-risk bias. This suggests that the studies included in this analysis did not exhibit serious publication bias.

### 3.3. Publication Offset Test

Egger regression coefficient test showed that all indicators had no published deviation (*p* < 0.05). The fail-safe N loss factor indicated minimal possibility of publication deviation. The fail-safe N value for ADG was 5097, which indicated that 5097 documents showing the opposite conclusions to a paper would be required to overturn the conclusions of the study (Table 1).

### 3.4. Effect of EO-Cr on the Growth Performance of Pigs

#### 3.4.1. Effect of EO-Cr on the Daily Gain of Pigs

Because of high heterogeneity in the literature results (*Q* = 9.712, *p* = 0.002), we selected the random effect model (Figure 2). The results showed that adding EO-Cr to pig feed significantly improved the ADG of weaned piglets (SMD = 1.498, *p* < 0.001) and growing-finishing pigs (SMD = 0.491, *p* < 0.001). The subgroup analysis results showed that various types of EO-Cr had different influences on the ADG of weaned piglets. The order of influence degree was as follows: chromium nicotinate (SMD = 2.928, *p* < 0.001) > chromium picolinate (SMD = 0.788, *p* < 0.001) > yeast chromium (SMD = 0.521, *p* = 0.162). However, the heterogeneity was still high after subgroup analysis (*Q* = 40.833, *p* < 0.001). Therefore, we concluded that the types of EO-Cr did not affect the heterogeneity of weaned piglets. According to the subgroup analysis results of growing-finishing pigs, the order of influence on the daily gain of growing-finishing pigs was as follows: chromium picolinate (SMD = 0.773, *p* = 0.042) > chromium methionine (SMD = 0.511, *p* = 0.007) > chromium propionate (SMD = 0.154, *p* = 0.232) > chromium yeast (SMD = 0.072, *p* = 0.852). Moreover, the heterogeneity was significantly reduced after subgroup analysis (*Q* = 4.396, *p* = 0.222), indicating that different types of EO-Cr led to high heterogeneity in growing-finishing pigs.

#### 3.4.2. Effect of EO-Cr on the Daily Feed Intake of Pigs

According to the effect of EO-Cr on the ADFI of weaned piglets and growing-finishing pigs (Figure 3), we used the random effect model to calculate that EO-Cr increased the ADFI of weaned piglets and growing-finishing pig, respectively. Thus, the effect of EO-Cr supplementation on the ADFI was significant. Similarly, according to the subgroup analysis of the impact of various EO-Cr forms on the ADFI of weaned piglets, yeast chromium showed the most significant impact on the ADFI of weaned piglets (SMD = 0.949, *p* < 0.001), whereas chromium picolinate (SMD = 0.651, *p* = 0.051) had the most significant impact on the ADFI of growing-finishing pigs. Following subgroup analysis, it was found that the heterogeneity was low (*Q* = 0.606, *p* = 0.436). This suggests that the pig’s growth stage, specifically the weaning stage, growth stage, and fattening stage, is a significant source of heterogeneity.

#### 3.4.3. Effect of EO-Cr on the Feed Gain Ratio of Pigs

Under conditions of high heterogeneity (*Q* = 5.848, *p* = 0.016), the use of organic chromium had a significant impact on the feed gain ratio (F/G) of weaned piglets (SMD = −1.179, *p* = 0.004) and improved their production levels (refer to Figure 4). However, there was no notable effect observed on growing-finishing pigs (SMD = −0.037, *p* = 0.878). In Figure 4, subgroup analysis was conducted to determine the effects of different types of organic chromium on the F/G of weaned piglets. The results showed that chromium nicotinate had the most significant effect on F/G reduction (SMD = −2.919, *p* < 0.001). Similarly, chromium picolinate was found to be the most effective organic chromium type in reducing F/G in growth-finishing pigs (SMD = −1.006, *p* = 0.049).

### 3.5. Effect of EO-Cr on the Carcass Quality of Pigs

Figure 5A shows a forest diagram of the overall impact of EO-Cr on the dressing percent of growing-finishing pigs and the corresponding subgroup analysis. The random effect model was selected to calculate the size of the effect. The overall effect was considerable (SMD = −0.002), but not significant (*p* = 0.989). Subgroup analysis showed that yeast chromium (SMD = 0.399, *p* = 0.484) had the greatest effect on the dressing percent of growing-finishing pigs; however, this effect was not significant. Furthermore, the heterogeneity test results after subgroup analysis were not significant (*Q* = 3.181, *p* = 0.365), which proves that the types of EO-Cr affected the level of heterogeneity.

According to Figure 5B, the random effect model improved the carcass lean ratio of growing-finishing pigs (SMD = 0.287) with statistical significance (*p* = 0.020). Subgroup analysis showed that chromium picolinate (SMD = 1.872, *p* = 0.198) dramatically impacted the carcass lean ratio of finishing pigs. According to the forest map, the sample number was small and the confidence interval range was large.

According to the forest diagram of the effect of EO-Cr on the back fat thickness of growing-finishing pigs (Figure 5C), the random effect model was selected for calculation because of the high heterogeneity. The results showed that EO-Cr reduced the back fat thickness of growing-finishing pigs (SMD = −0.407, *p* = 0.142). Subgroup analysis showed that chromium picolinate (SMD = 0.754, *p* = 0.335) had the greatest effect on back fat thickness. Furthermore, the form of EO-Cr induced heterogeneity (*Q* = 2.508, *p* = 0.474).

According to the random effect model, EO-Cr significantly increased the eye muscle area of growing-finishing pigs by SMD = 0.613 under high heterogeneity (*p* < 0.001) (Figure 5D). Subgroup analysis showed that chromium picolinate (SMD = 2.344, *p* < 0.001) had the most significant effect on the eye muscle area of growing-finishing pigs; this effect was statistically significant.

### 3.6. Meta-Regression Analysis

According to the regression analysis results in Table 2, the publication year and growth stage were sources of heterogeneity for the average daily gain, and the year of publication was a source of heterogeneity for the index of daily feed intake and dressing percent. The publication year, period of supplementation, and growth stage were not significant in the meta-regression analysis of F/G, carcass lean ratio, eye muscle area, back fat thickness, or other indicators, indicating that these variables were not sources of heterogeneity for these indicators.

### 3.7. Selection of Optimal EO-Cr Form for Pig Diet Supplementation

Few studies in the sampled literature added yeast chromium to the diet of weaned piglets, and we obtained no carcass quality index data for weaned piglets. Therefore, we only evaluated and analyzed the effects of chromium picolinate and chromium nicotinic acid on the growth performance of weaned piglets. We found that the addition of chromium nicotinic acid had the most significant impact on the growth performance of weaned piglets according to the size of the effect.

Additionally, few studies added chromium nicotinate to the diet of growing-finishing pigs; thus, studies of this type were not included. Only the effects of chromium picolinate, chromium propionate, chromium methionine, and chromium yeast on the growth and carcass quality of growing-finishing pigs were evaluated in this study. First, the membership degree was calculated (Table 3), and the fuzzy comprehensive evaluation results showed that chromium picolinate was the most suitable form of EO-Cr for supplementing the diet of growing-finishing pigs.

### 3.8. Optimal EO-Cr Supplementation Amount

Appendix A Figure A1 displays the correlation between EO-Cr supplementation dose and the effect size, growth performance, and carcass quality of pigs. The horizontal axis represents the SMD value, while the vertical axis represents the organic chromium dosage added in mg/kg. This curve allowed us to comprehensively predict the optimal dose of EO-Cr supplementation to the diets of weaned piglets and growing-finishing pigs. According to the growth performance of weaned piglets, the optimal dosage of chromium nicotinate was 0.125–0.150 mg/kg. However, because of a lack of research on high dosages, it is impossible to predict whether the growth performance of weaned piglets will be qualitatively changed at a higher dosage. The optimal dosage of chromium picolinate supplementation for growing-finishing pigs was 0.250–0.300 mg/kg.

## 4. Discussion

Existing literature has proved that Cr exerts different effects on pig growth, fattening, and carcass quality during different growth periods [47]. However, the potential influence of the type of EO-Cr supplementation had not previously been assessed. Here, we determined the effects of different types of EO-Cr on the growth, fattening, and carcass quality of pigs at different growth stages by meta-analysis. Specifically, we tested deviations in the published articles used in this meta-analysis using Egger’s regression coefficient and a loss of safety coefficient. Generally speaking, publication bias is caused by the fact that in similar studies, studies with statistically significant results are more likely to be accepted and published than those without statistical significance. In order to prevent this, all relevant studies should be included as far as possible. From the results, all indicators in this paper have no publication bias, indicating that the articles included in this study are relatively comprehensive and exhibit no more positive conclusions than negative ones. Thus, the results are scientifically valid.

The average daily gain is an essential measure of the growth performance of pigs, which is affected by multiple factors, such as genetic breed, animal diet, and farm management [27]. Our meta-analysis found that supplementation of EO-Cr in pig diets significantly improves the ADG of weaned piglets and growing-finishing pigs, which is consistent with previous research [33]. This finding is attributed to EO-Cr supplementation increasing the mRNA levels of growth hormone in serum and promoting growth hormone secretion [4].

The main role of Cr is to facilitate the binding of insulin and its receptors on the cell membrane [48]. Insulin sensitive cells in turn capture more glucose, which is converted into a greater amount of energy, with the additional energy used for increasing lean gain and reducing fat contents in the carcass as adipose tissue deposition is less efficient than muscle deposition [49]. Thus, dietary supplementation of EO-Cr for pigs will improve feed efficiency and reduce feed costs [50].

Moreover, Cr is an active component of the glucose tolerance factor, which increases cell membrane fluidity, potentializes insulin action, promotes insulin binding to its receptors, enhances cell sensitivity to glucose, and accelerates the conversion of thyroxine to triiodothyronine [51]. Therefore, Cr is indispensable for the metabolism of carbohydrates, lipids, and proteins as an essential compound in the formation of the glucose tolerance factor. Moreover, Cr increases the content of insulin-like growth factor-I, which plays a key role in the growth rate of pigs and the utilization of nutrients [52].

In pasture production, to obtain high-quality pork, it is sometimes necessary to control the feed intake of pigs to reduce their weight gain and prevent excessive fat accumulation; such action simultaneously reduces production costs [53]. Our results show that adding EO-Cr does not cause a rapid increase in the daily feed intake of weaned piglets or growing-finishing pigs; therefore, it does not cause excessive fat accumulation but maintains a slightly high feed intake. This phenomenon is caused by the fact that EO-Cr can enhance the tolerance of the animal body to glucose, stimulate glucose uptake by tissues, and accelerate the clearance rate of glucose in the blood, thereby reducing the level of glucose in the blood and promoting food intake [54].

The feed–gain ratio is an important parameter for measuring the feed reward and feed conversion rate. Our analysis results show that adding EO-Cr to pig diets can significantly reduce the feed–gain ratio of weaned piglets, which is consistent with Xiao Xu’s research results [41], but has no significant impact on growing-finishing pigs. The observed feed–gain ratio reduction may be caused by reduced food consumption or increased meat production. Therefore, we proved that adding EO-Cr to feed can improve production efficiency, feed conversion rate, and remuneration, and therefore increase the economic benefits.

The carcass quality of pigs is closely related to the quality of pork. The results of this study show that the dressing percent of carcass indexes can be slightly improved after adding EO-Cr to the diet of growing-finishing pigs; however, this effect is not significant. This indicates that adding EO-Cr cannot significantly increase the carcass dressing percent or back fat thickness of growing-finishing pigs. Conversely, regulation of the carcass lean ratio and eye muscle area is significant, which indicates that adding EO-Cr can significantly improve the carcass lean ratio and eye muscle area of growing-finishing pigs. This is because supplementation with EO-Cr can promote fat decomposition and protein synthesis, reduce the level of circulating cholesterol in the blood, promote the synthesis of cortisol, and improve carcass and meat quality. These findings are consistent with those of previous literature [23,30,31,41].

Overall, the effect of EO-Cr supplementation on pig growth performance is much more significant for weaned piglets than growing-finishing pigs, which is related to the unique physiological stage of weaned piglets. Therefore, adding EO-Cr to feed at the weaned piglet stage can maximize the growth performance. According to the prediction results of the effective dose curve, the optimal growth performance index can be achieved by supplementing 0.125–0.150 mg/kg of EO-Cr in the feed-weaned piglets. Furthermore, 0.250–0.300 mg/kg is the optimal amount of EO-Cr supplementation for growing-finishing pigs. 

The sources of heterogeneity in the studied literature can be attributed to three aspects. (1) Heterogeneity in the year of publication is attributed to significant improvements in EO-Cr supplementation and the measurement of indicators with more advanced test conditions; this means that the measurement values in early studies differ from those in more recent research. (2) Heterogeneity in the growth stage, i.e., weaned stage and growth and fattening stage, was reduced by subgroup analysis for most indicators. (3) Heterogeneity in the period of supplementation, which refers to the duration of EO-Cr supplementation, has a significant effect on the daily gain.

## 5. Conclusions

In conclusion, we evaluated the effects of EO-Cr supplementation on the growth performance and carcass quality of pigs via a meta-analysis. Dietary supplementation of EO-Cr significantly increased the ADG (*p* < 0.001) and daily feed intake (*p* < 0.001). For growing-finishing pigs, EO-Cr significantly increased the ADG (*p* < 0.001), ADFI (*p* = 0.002), lean ratio (*p* < 0.001), and eye muscle area (*p* < 0.001), but had no significant effect on the feed–gain ratio (*p* = 0.112), dressing percent (*p* = 0.988), or back fat thickness (*p* = 0.210). Furthermore, the effect of adding EO-Cr to pig diets was greater for weaned piglets than for growing-finishing pigs. The optimal types of EO-Cr for weaned piglets and growing-finishing pigs was chromium nicotinate at a dosage of 0.125–0.150 mg/kg and chromium picolinate at a dosage of 0.250–0.300 mg/kg, respectively.

## Figures and Tables

**Figure 1 animals-13-02014-f001:**
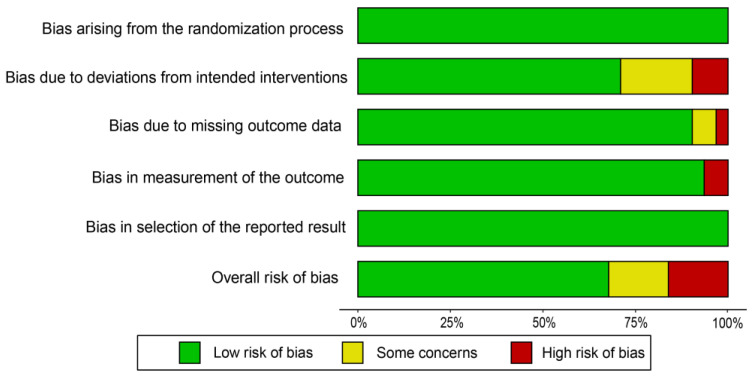
Study quality assessment.

**Figure 2 animals-13-02014-f002:**
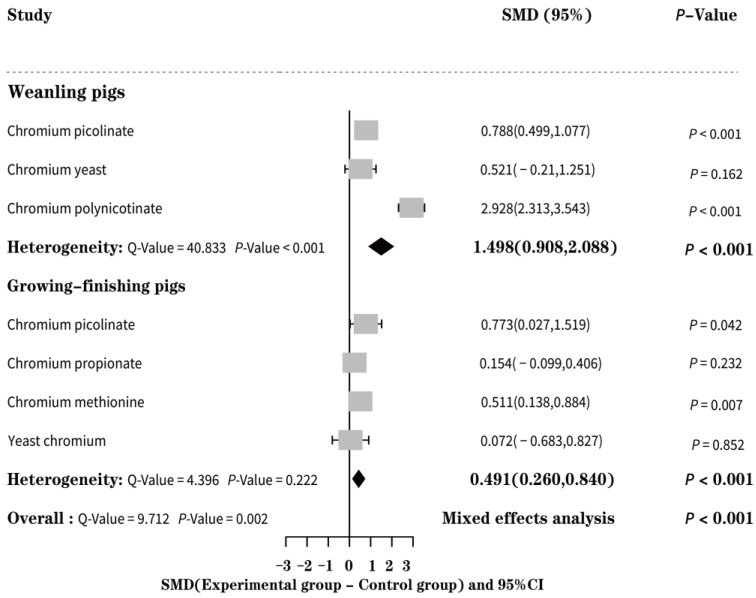
Forest effect of EO-Cr supplementation on the average daily gain of weaned and growing-finishing pigs.

**Figure 3 animals-13-02014-f003:**
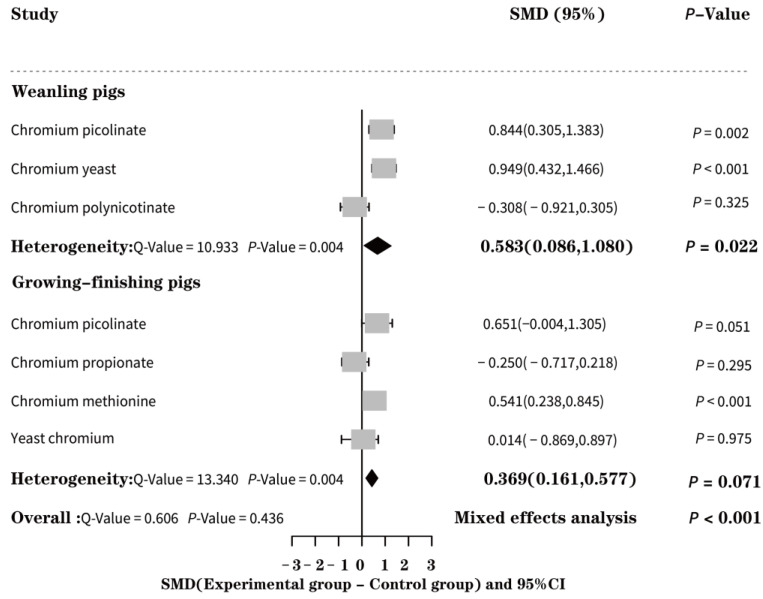
Forest effect of EO-Cr supplementation on the average daily feed intake of weaned and growing-finishing pigs.

**Figure 4 animals-13-02014-f004:**
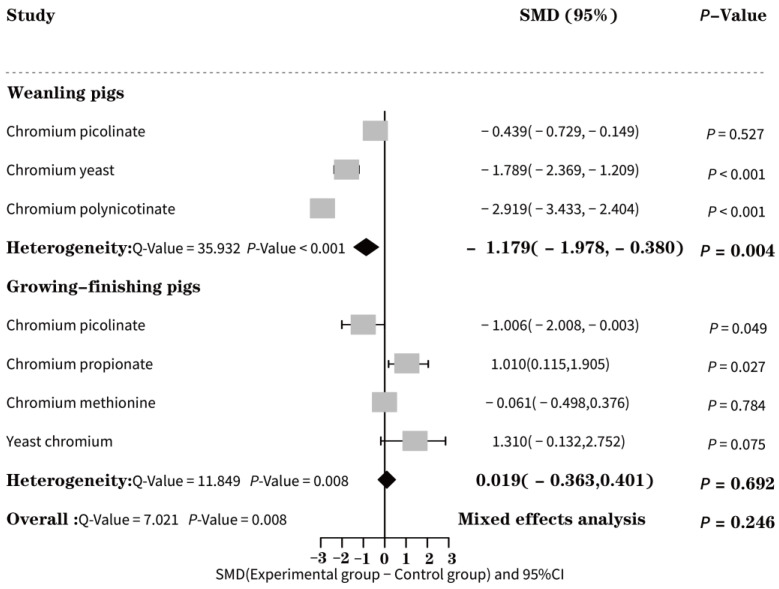
Forest effect of EO-Cr supplementation on the feed gain ratio of weaned and growing-finishing pigs.

**Figure 5 animals-13-02014-f005:**
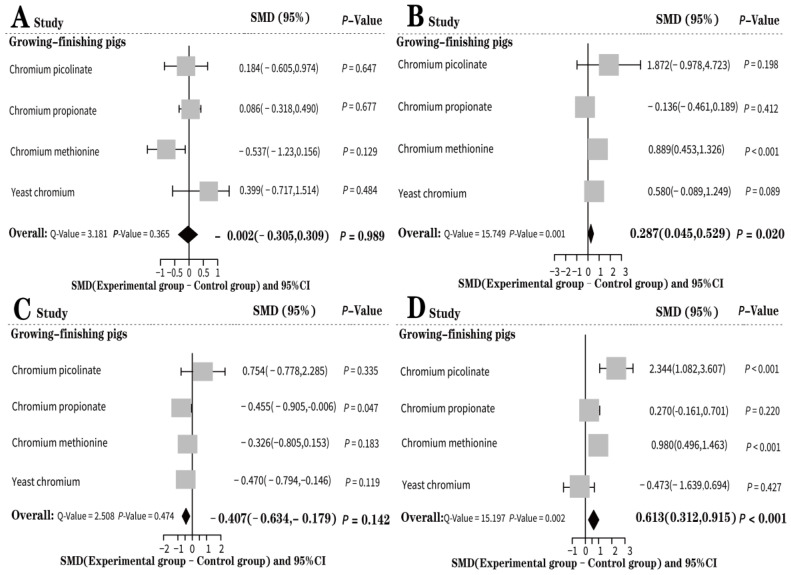
Forest effect of EO-Cr supplementation on the carcass quality of growing-finishing pigs. (**A**): Effect of adding different EO-Cr forms on the dressing percent of growing-finishing pigs; (**B**): effect of different EO-Cr forms on the carcass lean ratio of growing-finishing pigs; (**C**): effect of different EO-Cr forms on the back fat thickness of growing-finishing pigs; (**D**): effect of adding different EO-Cr forms on the eye muscle area of growing-finishing pigs. Number of treatment methods (study), Values are the mean and standard deviation (mean, SD), the standardized mean difference (SMD), and the 95% confidence interval (95%). Test results of heterogeneity and overall addition of EO-Cr are listed in the bottom left.

**Table 1 animals-13-02014-t001:** The deviation test was carried out on the average daily gain (ADG), average daily feed intake (ADFI), feed gain ratio (F/G), slaughtering volume, carcass lean ratio, eye muscle area, and back fat thickness. Two test methods were used in this study; the first is Egger test, and the second is fail-safe N, Nfs test.

Indicators ^1^	P-Egger ^2^	Fail-Safe N, Nfs ^3^
ADG	0.11	5097
ADFI	0.26	2155
F/G	0.57	275
Dressing percent	0.86	1237
Carcass Lean Ratio	0.15	937
Back fat thickness	0.97	442
Eye muscle area	0.10	2171

^1^ publication offset test indicators. ^2^ indicates a significant publication bias (*p* < 0.10). ^3^ indicators used to assess publication bias.

**Table 2 animals-13-02014-t002:** After meta-analysis of the average daily gain (ADG), average daily feed intake (ADFI), feed gain ratio (F/G), dressing percent, carcass lean ratio, eye muscle area, and back fat thickness, the meta-regression analysis was conducted to explore whether the publication year, period of supplementation, and growth stage were the sources of high heterogeneity.

Indicators ^1^	Meta-Regression Analysis		
	Publication Year ^2^	Period of Supplementation ^3^	Growth Stage ^4^
ADG	0.038 *	0.203	<0.001 ***
ADFI	0.022 *	0.329	0.638
F/G	0.073	0.419	0.053
Dressing percent	0.015 *	0.751	NS
Carcass Lean Ratio	0.491	0.422	NS
Back fat thickness	0.059	0.332	NS
Eye muscle area	0.471	0.961	NS

^1^ Various indicators of meta-regression. ^2^ Year of publication of the research article. ^3^ Duration of organic Cr supplementation. ^4^ Effect of growth stage on the index of organic Cr supplementation. * *p*-value of *t*-test in meta-regression analysis. *p* < 0.05 *, *p* < 0.01 **, *p* < 0.001 ***. NS indicates no data.

**Table 3 animals-13-02014-t003:** The membership function obtained from the four significant indicators of average daily gain (ADG), average daily feed intake (ADFI), carcass lean ratio, and eye muscle area is added and calculated. The results reflect the comprehensive evaluation scores of four different organic chromium after dimensionless.

Types of organic Cr ^1^	ADG ^2^	ADFI ^3^	Carcass Lean Ratio ^4^	Eye Muscle Area ^5^	Summary ^6^
Chromium picolinate	1.000	1.000	1.000	1.000	4.000
Chromium propionate	0.514	−0.384	0.044	0.158	0.331
Chromium methionine	0.661	0.831	0.475	0.418	2.385
Yeast chromium	0.093	0.022	0.310	−0.202	0.223

^1^ Types of organic chromium included in the membership function. ^2^ Membership degree of average daily weight gain. ^3^ Membership degree of average daily feed intake. ^4^ Membership degree of carcass lean ratio. ^5^ Membership degree of eye muscle area. ^6^ Total membership degree.

## Data Availability

The datasets used and analyzed during the current study are available from the corresponding author on reasonable request.

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
