# Peer review of "Meta-Analysis of the Effects of Organic Chromium Supplementation on the Growth Performance and Carcass Quality of Weaned and Growing-Finishing Pigs"

_animals, 2023, doi:10.3390/ani13122014_

Round 1
Reviewer 1 Report
General comments:
The authors need to define “organic chromium”. This could be interpreted to be an ingredient that is produced under an organic certification scheme.
The authors need to improve consistency of terminology throughout the manuscript. Use consistent terms when referring to items. For example, Feed Gain ratio vs. feed meat ratio or growing-finishing vs. growing-fattening, etc.
Matching data in the text to data presented in the tables and figures is difficult in many spots. The authors need to be more transparent about where the text data comes from.
Specific comments:
L22: It is not clear what “test cycles” refers to. This is an awkward term. The text would be clearer if the authors used “feeding period” or “period of supplementation” or something similar.
L31-32: How was this effect determined statistically? It does not appear that there was any statistical evaluation of this statement.
L53-54: “…more easily absorbed with absorption efficiency.” This sentence is awkward.
L56-58: This statement is puzzling. What is the point being made here?
L84-89: It seems that an additional inclusion criterion should be that the papers reported the same outcomes from feeding chromium.
L108-115: Did the authors act on this research quality assessment? The assessment was conducted and results reported but this is the Materials & Methods section so it seems that there should have been some adjustments to improve the quality of the input data.
L157-158: What does this statement mean? Was there missing data? How much was missing? What does this mean for the conduct of the study?
L162-166: What P value threshold was used to deem a response is significant?
L170-171: Just eliminating duplicate studies reduced the pool from 5977 to 35 studies? It seems there were other criteria in play to remove over 5900 studies.
L178: The conclusion of what study? Conclusion of the current review?
L179 (Table 1): “Slaughter rate” and “Lean meat rate” need to be defined. These are not common terms in the swine industry. Do the authors mean “Dressing percent” and “Carcass Lean”?
L215: Feed Meat Ratio is different than Feed Gain ratio. Be consistent. Gain includes more than just meat.
L221: Figure 4 states chromium polynicotinate. The authors need to be consistent throughout text and tables and figures. Likewise, the text refers to growing-finishing pigs but Figure 4 lists growing-fattening pigs. I know they are the same class of pig but consistency makes the paper easier to understand.
L232: Consistency again. Feed gain ratio instead of feed weight ratio
L238: A SMD of -0.002 does not seem all that “substantial”
L244 & 245: These P values do not match those listed in Fig. 5.
L243-253: The statistical references in this text do not show up in Figure 5. Likewise, there is no indication of how many replications are included in data presented in Fig. 5.
L252-253: The greatest effect on backfat was 0.754 from chromium picolinate. This effect is larger than yeast chromium albeit in an unfavorable direction.
L267: what is a presidential measurement?
L282: Footnote 4 does not appear in the body of Table 2
L288-290: Where are the data to support this statement?
L301 (table 3): Define the summary column heading
L303: There is no Figure 6 in this paper.
L309-312: It is not at all clear to this reviewer how the authors determined these optimal dosages.
L350-353: This statement seems very peculiar and out of place here. What is the point of this sentence? How does this sentence in this location contribute to your story?
L360-367: It seems that authors are confounding weight gain with meat production. The two are different.
L376-378: The authors did not measure these traits. This text is written as if they did.
L390-393: What is the meaning of this text?
L397: The meaning of this “stable state” after chromium is stopped is not clear. What is the point here?
L405-411: This text is summary not conclusions.
Table A1:
· Need to define the column heading “number”
Table A2:
· What does N represent? Number of pigs? Number of pens? What?
· Serial 22 Lindemann does not appear in the Lit. Cited section
Figure A1:
· This figure is very difficult to interpret. What do the legends mean?
· Is “Wight” supposed to be “Weight”?
Terminology needs to be changed to match those terms typically used in swine production or define more clearly the terms used. For instance, slaughter rate, what does it mean?
Reviewer 2 Report
The authors should distinguish the different forms of Chrome. All of them are not reported to be toxic. They should as well differentiate atom and ions.
Line 101: within group SDs could not be an estimate of SD, SEM, residual SD... (in my opinion)
Line 169-172: the skip from 5977 to 35 should be better explained.
As a rule, a Conclusion section should not contain results or statisticial significance indications.
Figure A1. Check the Legends (it semms not to have a link between the points sizes and n. Check the title. Change Wight to Weight. Are curve lines useful?
As far as can be seen, there are no mention of the contents of Cr-ion in any control diet.
